# Comorbidity of Post-Traumatic Stress Disorder and Alcohol Use Disorder: Animal Models and Associated Neurocircuitry

**DOI:** 10.3390/ijms24010388

**Published:** 2022-12-26

**Authors:** Bo Zhan, Yingxin Zhu, Jianxun Xia, Wenfu Li, Ying Tang, Anju Beesetty, Jiang-Hong Ye, Rao Fu

**Affiliations:** 1Department of Anatomy, School of Medicine, Shenzhen Campus of Sun Yat-sen University, Sun Yat-sen University, Shenzhen 518107, China; 2Department of Basic Medical Sciences, Yunkang School of Medicine and Health, Nanfang College, Guangzhou 510970, China; 3Department of Biology, School of Life Science, Southern University of Science and Technology, Shenzhen 518055, China; 4Department of Anesthesiology, Pharmacology, Physiology & Neuroscience, Rutgers, New Jersey Medical School, The State University of New Jersey, Newark, NJ 07103, USA

**Keywords:** comorbid PTSD and AUD, animal model, ventral tegmental area, mesolimbic reward circuit, hippocampus, amygdala, dopamine, the prefrontal cortex, paraventricular nuclei, locus coeruleus

## Abstract

Post-traumatic stress disorder (PTSD) and alcohol use disorder (AUD) are prevalent neuropsychiatric disorders and frequently co-occur concomitantly. Individuals suffering from this dual diagnosis often exhibit increased symptom severity and poorer treatment outcomes than those with only one of these diseases. Lacking standard preclinical models limited the exploration of neurobiological mechanisms underlying PTSD and AUD comorbidity. In this review, we summarize well-accepted preclinical model paradigms and criteria for developing successful models of comorbidity. We also outline how PTSD and AUD affect each other bidirectionally in the nervous nuclei have been heatedly discussed recently. We hope to provide potential recommendations for future research.

## 1. Introduction

Post-traumatic stress disorder (PTSD) is a maladaptive and debilitating psychiatric disorder. Symptoms, such as re-experiencing, avoidance, negative emotions and thoughts, and hyperarousal in the months and years following exposure to severe traumatic events, are the characteristics of PTSD [1]. According to DSM-5 diagnostic criteria, PTSD patients are typically subject to the traumatic event (Criteria A) accompanied by the following core symptoms over one month: Intrusive memories (Criteria B); Avoidance behaviors (Criteria C); Negative mood and cognition (Criteria D); Arousal and sleep disturbances (Criteria E) [2].

Alcohol use disorder (AUD) is a chronic brain disease involving the transition from recreational drinking to binge or heavy drinking to high levels of intoxication, leading to compulsive intake, losing control in limiting intake, and a negative emotional state when alcohol is removed and constant motivation to seek and to consume alcohol despite adverse consequences [3,4].

AUD and PTSD often reciprocally interact [5]. People with PTSD are more prone to develop drinking problems, with nearly 1.2 times higher incidence than those without PTSD [6]. Up to 75% of people who suffer severe violent, traumatic events report drinking problems. Severe trauma associated with negative emotion, impulsivity, and mood lability, potentially drives drinking. About 60–80% of Vietnam Veterans seeking PTSD intervention have reported alcohol use problems. PTSD veterans over 65 have a higher risk of committing suicide once they have a comorbidity of AUD and depression [7]. This information implies that PTSD facilitates AUD development. PTSD patients with depression or anxiety are more likely to drink alcohol to alleviate affective disorders [8]. Repeated alcohol consumption for temporarily mitigating PTSD symptomatology leaves the patients more susceptible to AUD [9]. 

Conversely, AUD patients have a higher incidence rate of PTSD than non-drinkers. In a study of 1098 African American drug users out for treatment, 44% of the participants were diagnosed with PTSD [10]. Moreover, the early onset of alcohol addiction is associated with an increased risk for traumatic event exposure in both genders. This is likely because when intoxicated, AUD patients’ ability to detect dangerous environmental cues is impaired, increasing the possibility of trauma exposure. AUD patients with binge drinking and intoxication are likely to put themselves in high-risk situations [11]. Likewise, alcohol abuse makes achieving self-recovery from exposure to traumatic events more difficult, thereby promoting PTSD development [12].

Although PTSD and AUD mutually contribute to the development of each other, PTSD patients with concomitant AUD are more common and deserve further detailed discussion. Indeed, there is a more significant number of PTSD patients secondarily diagnosed with AUD than AUD patients accompanied by PTSD [5]. Patients with PTSD and AUD comorbidity improved symptoms of both disorders, while those with alleviated AUD symptoms did not necessarily witness PTSD improvement [13].

Despite the high incidence of comorbidity, there is a significant gap in understanding how stress and traumatic experience interact with the three-stage cycle framework, including binge/intoxication, withdrawal/negative affect, and craving for alcohol addiction. 

We conducted a comprehensive search on PubMed databases and PsycINFO using Medical Subject Headings terms, including “post-traumatic stress disorder”, “alcohol addiction”, and “neural circuits” in various combinations to retrieve literature that introduced animal models for the comorbidity of PTSD and AUD. We used similar strategies to identify articles on the neurobiology of comorbidity. The inclusion and exclusion criteria for the articles were kept flexible. We expanded the scope of the review based on findings from the study of essential papers and reports.

This review summarizes recent advantages in the animal models of PTSD and AUD comorbidity research. We then analyze neurocircuitry critical for developing and maintaining PTSD and AUD, particularly those shared by both disorders, such as alternations in neurotransmitters and stress systems.

## 2. Preclinical Models of the PTSD-AUD Comorbidity

### 2.1. Preclinical Models for PTSD

Animal models are an effective experimental tool for mimicking human diseases. Given that stress and traumatic experiences are necessary to induce PTSD, the typical paradigms precipitate the disorder by applying physical, social, and psychological stressors individually or in combination imposed on rodents to develop disorders. Moreover, by manipulating stress type, intensity, duration, and frequency, the preclinical models reflect core PTSD phenotypes measured through various behavioral assays. 

The rodents are heavily used as animal models for PTSD studies. Up to the end of 2022, over 1300 studies used rats, and approximately 700 used mice. Caution is needed when interpreting results since significant differences regarding morphological structures, stress responses, physiologies, genetics, and behaviors exist between the two species [14]. Over the past two decades, more mice were utilized in studies to dissect mechanisms related to immunology or genetics (Table 1). It’s worth noting that epidemiological assessments of gender have emphasized the importance of studying the susceptibility of females, female traumas, and sexual abuse. For example, in rats exposed to predators’ odor, females were more likely to exhibit PTSD-like phenotype, while female models were hard to establish when exposed to social defeat [15].

The earliest animal model for PTSD could be dated back to the “learned helplessness” paradigm, which was based on Pavlov’s “classical conditioning” [16], in which physical stressors are mainly used to develop PTSD models. Those stressors include electric shock, underwater trauma, restraint/immobilization stress, and single prolonged stress (SPS). 

After a certain degree of electric shock, the animals exhibit behavioral responses like human PTSD intrusion symptoms, such as crouching against the chamber wall, increased respiratory rate, and prolonged freezing time when placed in an environment similar to an electric shock chamber [17].Uncontrollable and unpredictable foot shock through their feet or tails is the most common stress exposure, such that even a single exposure can lead to long-lasting behavioral changes [18].

In contrast to the physical stressors that apply well-defined and precise traumatic experiences, social defeat and predator stresses as socio-psychological stressors focus on ecological validity to increase translatability between animal models and humans. Subsequently, a series of behavioral tests were used to assess the expected outcomes of PTSD. These measurements include: (1) contextual and cued trauma reminders for intrusion; (2) elevated zero/plus/T-maze, light-dark box, conditioned object/taste aversion, and open field for avoidance; (3) Morris water/Barnes maze, radial arm water maze, anhedonia, forced swimming test, novel object recognition, social preference/avoidance test, response bias probabilistic reward task, step-through inhibitory avoidance for mood and cognition; and (4) object/marble burying, startle response, open field, aggression, electroencephalogram, prepulse inhibition, operant attentional set-shifting tasks, etc. [19]. 

Three criteria are frequently induced to assess the eligibility of PTSD animal models: face validity, construct validity, and predictive validity [20]. Face validity means the animal model should show similar PTSD-like symptoms to those detected in humans, especially the four core symptoms listed above. Construct validity is a theoretical rationale. Since the cause (traumatic experiences) is relatively straightforward, a model with a similar construct (using trauma exposure) can be developed. Moreover, predictive validity shows whether the model can predict performance, in which how well the test works is established by measuring it against known criteria. These predictable outcomes may be treatments or behavioral or physiological markers of the disorder (Table 2).

### 2.2. Preclinical Models for AUD

Essential characteristics of alcohol consumption have been found in many mammal species, fish, birds, and non-spinal animals [21]. However, significant neuropharmacology and brain circuitry differences between non-mammalian model systems and mammals limit the use of animals mentioned above. Rodents are the most commonly used animal models.

To establish AUD animal models, we took advantage of ethanol’s reinforcing property. Training rodents to drink ethanol under the intermittent access 2-bottle free choice (IA2BC) drinking paradigm in homecage is a well-established paradigm to induce escalation of consumption in either low drinking species (Sprague Dawley) or moderate (Long-Evans and Wistar), as well as high drinking species (P-rats) [22]. The IA2BC paradigm could induce dependence since affective disorders, including anxiety-, depression-like behavior, and pain was observed during acute abstinence [23,24]. In addition, researchers sometimes mixed sweeteners with alcohol to achieve quicker intake escalation in rodents [25].

Conversely, researchers could further differentiate the compulsivity by adding the aversive stimuli factors, like mixing quinine and conjugating a foot shock followed by active lever pressing before the alcohol is delivered [26]. The alcohol self-administration model examines the animal’s motivation for ethanol. The IA2BC was arranged ahead of operant SA to habituate the drinking pattern of rats [24,27].

As for passive alcohol consumption, many studies established an alcohol dependence using intermittent alcohol vapor [27,28]. The application of alcohol vapor can lead to blood alcohol levels in rodents comparable to human alcoholic patients [29]. Intermittent alcohol vapor exposure makes animals experience multiple withdrawal episodes, which accelerates alcohol dependence [30]. Animals in the intermittent alcohol vapor paradigm consumed more alcohol than those in the continuous alcohol vapor [31]. Intragastric infusions and intraperitoneal ethanol injection are also methods of passive ethanol consumption [21,32].

Alcohol-related studies’ most commonly used behavioral assays concentrated on binge drinking, withdrawal-associated adverse affective disorders, and craving behaviors [33]. The behavioral tests are summarized according to the test’s purpose (Table 3).

Acute ethanol exposure could lead to behavioral sensitization reflected by locomotor activity enhancement after intoxication [34]. The open field test is mostly used to detect changes in animal activity by recording the motion trail and the time spent in the central and surrounding areas. The conditioned place preference (CPP) was also used to test the reinforcing property [35]. 

In clinical studies, the stimulation of motor function by low-dose alcohol is associated with an increased risk for a lifetime diagnosis of alcohol dependence [36]. In rodents, ethanol-induced behavioral sensitization and loss of the righting reflex (LORR) [37] were used to predict and screen rodents with possible high rewarding effects [38]. 

The development of drug abuse usually depends on the balance between its rewarding and aversive properties. Therefore, the aversive effect of alcohol should not be neglectable [39]. Conditioned place aversion (CPA) and conditioned taste aversion (CTA) are widely used to assess the aversive property of addictive drugs by documenting times spent in the different compartments [40] and comparing the amount of saccharin consumed in rodents injected with various doses of ethanol, respectively [39].

Negative affect is an essential feature of alcohol withdrawal, including elevated anxiety levels, attenuated ability to experience pleasure from natural rewards, and depression. The marble burying test assesses the anxiety level of rodents. The decreased consumption of natural rewards reflects anhedonia. At the same time, the prolonged immobility in the forced swimming test indicates a diminished will to survive and depression [41] (but may show an adaptation).

Craving is a characteristic of AUD. Thus, assessing active lever pressing to acquire alcohol in dependent rats in the operant chamber could help evaluate their alcohol-seeking behavior. Drinking in the dark is a paradigm used to assess mice’s binge drinking behavior. IA2BC paradigms are often used to train rodents to consume alcohol in their home cages.

Alcohol self-administration is used to evaluate rodents’ alcohol-seeking behavior [42]. Rodents keep pressing the lever for alcohol despite knowing they are likely to suffer a foot shock. The “breakpoint” that the lever being touched reflects the intensity of ethanol-seeking motivation. The punishment-imposed ethanol-lever press is widely used to evaluate compulsivity [42]. 

AUD animal models have limitations. Firstly, many factors could influence animals’ susceptibility to develop alcohol addiction. Additionally, AUD patients have mental and physical phenotypes, but in animal models, we can only observe mental changes through behavioral tests, limiting the overall understanding of the disease. 

### 2.3. Preclinical Models for PTSD Comorbid with AUD

However, compared to numerous PTSD or AUD preclinical studies, the literature coupling the two is sparse, and few animal models are being reported. There are many reasons for this. The most important one may be the complexity of combining the models used to study each disorder [43]. It is well-accepted that stress triggers negative emotional states and subsequent adaptive changes that lead to AUD development. Thus, external stressors were usually proposed to establish an animal model of PTSD, followed by a battle of behavioral testing responses to acute and chronic effects of ethanol in affected animals. When using this combination strategy, a critical consideration is whether stress is sufficient to cause trauma to the animal. Secondly, whether an escalation of alcohol-drinking-related behaviors could be induced should be a necessary component of the comorbidity model. For example, well-accepted alcohol-dependent animals are characterized by the following features. (1) Orally self-administer ethanol. (2) the amount of ethanol consumed should result in pharmacologically relevant BACs. (3) the animal should consume ethanol for its post-ingestive pharmacological effects and not strictly for its caloric value or taste. (4) Ethanol should be positively reinforcing, so animals will work to access ethanol. (5) Chronic ethanol consumption should lead to metabolic and functional tolerance expression. (6) Chronic ethanol consumption leads to dependence, as indicated by withdrawal symptoms after access to ethanol is terminated. (7) the animals should express relapse-like drinking behavior, which manifests as a loss of control [44,45]. (8) Animals might display binge-like drinking and the expression of excessive ethanol consumption during the juvenile, adolescent, and emerging adult stages of development (e.g., [46]). Here, we introduce some details of basic models and their relevance to comorbidity.

The Single Prolonged Stress model is a translational comorbid condition model. It involves a single prolonged intense stress exposure combined with three stressors. Such a model can generate behavioral symptoms associated with PTSD, including reduced fear extinction, enhanced startle response, hyperarousal, increased anxiety-like behavior, and an escalation of ethanol consumption indicative of AUD [47].

Social isolation, such as maternal deprivation, is a demonstrated risk factor for alcohol consumption during adolescence and adulthood. Some sets of animal models aimed at understanding the effect of traumatic stress on future alcohol drinking have adapted an early life stress model to show that social isolation during adolescence, but not during adulthood, increases alcohol drinking. Socially isolated rats displayed hyperactivity in a novel environment, enduring increases in alcohol intake and preference, increased anxiety-like behavior, and notably, impaired fear extinction [48]. These behaviors are all associated with increased vulnerability to AUD and PTSD.

Another popular model involves exposing animals to the odor of a natural predator to mimic a life-threatening situation without causing actual harm to the animal [49]. In such models, adult male rats were trained to drink dessert wine in their cages before encountering dirty cat litter. Rodents that displayed high reactivity to the predator odor display behavioral phenotypes associated with alcohol dependence, including higher alcohol drinking, compulsive-like responding, and extreme behavioral response. This model is unique in its replication of the avoidance symptom of PTSD [50].

The establishment of a comorbidity model is determined by how these two disorders couple. Methods used in animal studies addressing comorbidity are based on experimental aims or interests. For example, McCool and Chappell reported postweaning social isolation increased the expression of anxiety-like behavior in the elevated plus maze. Ethanol consumption was also increased during continuous home-cage access with the 2-bottle choice paradigm. During operant self-administration, isolation housing increased the response rate and increased ethanol consumption [51]. It has also been reported that excessive drinking increases the risk of PTSD development. For example, Holmes and colleagues demonstrated that chronic intermittent ethanol (CIE) might increase the risk for trauma-related anxiety disorders by disrupting the mPFC-mediated extinction of fear [52].

Specific comorbidity designs should be adjusted, including the stressor’s nature, alcohol exposure patterns, and the sequence of the two interventions, according to the particular purpose of the experiment. We proposed that since the interpretation of behavioral characterization and phenotyping are essential to test hypotheses and evaluate potential pharmacological agents for PTSD, the following factors should not be neglected. Some methods employed for comprehensive behavior combined with pharmacological tests are complex. In addition, because some tests have interactions, a single test might reflect multiple PTSD symptoms. Therefore, it must be very cautious about arranging the order of behavioral tests and data interpretation, ensuring the translational behavioral result are robust, reliable, and reproducible.

## 3. Neurocircuitry of the PTSD-AUD Comorbidity

Testing and developing theoretical hypotheses of PTSD-AUD comorbidity requires specific data on molecular mechanisms. Most recent advances in neural circuitry in PTSD and AUD have been derived from the specific preclinical models described above. These animal models can be used to study specific elements of the disease process, laying the foundation for learning neural circuits. Here, we focus on the potential brain regions or neural circuitry involved in comorbid PTSD and AUD from the perspective of a single lesion and the symbolic overlap of the two diseases. The ventral tegmental area (VTA), hippocampus, amygdala, prefrontal cortex (PFC), paraventricular nuclei (PVN), and locus coeruleus (LC), are well-associated with processing fear, anxiety, stress, and rewards (See Table 4).

### 3.1. VTA

The VTA in the midbrain is enriched with dopaminergic (DA) neurons that project to limbic regions through the mesolimbic pathway [78]. The limbic system, comprised of the VTA, the nucleus accumbens (NAc), and the hippocampus, is responsible for incentive salience, decision-making, working memory, reward, and aversion [79]. 

Since stress or traumatic experiences are necessary to induce PTSD, various paradigms are used to precipitate the disorder by applying physical, social, and psychological stressors individually or collectively. Stress affects DA neuron activity and elevates dopamine levels in the mesolimbic system. Acute stressful situations have a stimulating effect on VTA-DA neurons. At the same time, chronic stress exerts an allostatic shift in the mesolimbic DA system to be hypoactive and hyporesponsive in the long term [54]. Alterations in mesolimbic DA neurotransmission allow behavioral adaptations in response to various environmental stimuli. They, thus, are essential for stress coping. 

The effect of acute stress on the VTA-DA neurons is short lasting and characterized by initial transient activation. Acute stress can expedite the initial stage of AUD. Stressful stimuli could increase the baseline VTA dopamine level, sensitize the rewarding circuits, and potentially enhances VTA-DA neuron burst firings. Thus, acute stress can enhance ethanol’s rewarding property and facilitate binge drinking behavior [80]. In addition, acute psychological stressors can increase extracellular dopamine levels in regions innervated by VTA neuronal projections, such as NAc [36,53]. It is well accepted that the mesolimbic DA system can become hyper-reactive in response to drug reward, which is one of the characteristics of drug abuse, including alcohol addiction [81,82].

There are several reasons accounting for the phenomenon mentioned above. Specifically, acute stress can produce excitatory synapse proliferation in VTA, broadly promoting the DA output. For example, glutamatergic receptors, including α-amino-3-hydroxy-5-methyl-4-isoxazole propionic acid receptors (AMPARs) and N-methyl D-aspartate receptors (NMDARs), are ubiquitously expressed in the area receiving inputs from VTA-DA neurons. Besides, inhibitory synaptic transmission in the VTA is also susceptible to undergoing plastic changes during acute stress [54]. Such cellular configuration can stimulate mesolimbic DA release in the VTA. In conclusion, with the effect of stress, the sensitized VTA DA neurons significantly increase the rewarding property of alcohol, which increases the propensity to develop alcohol abuse.

Such cellular configuration can stimulate mesolimbic dopamine release. In conclusion, with the effect of stress, the sensitized VTA DA neurons significantly increase the rewarding property of alcohol, which increases the propensity to develop alcohol abuse. 

Morphological changes in VTA DA neurons are associated with depressive symptoms. Moreover, anhedonia, one of the core features of depression [83], contributes to problematic alcohol use [84]. Anhedonia prompts PTSD patients to self-medicate alcohol to alleviate negative moods, probably by compensating for low dopamine levels. It is well-accepted that anhedonia can significantly increase relapse drinking and reduce treatment efficacy [85].

VTA Gamma-aminobutyric acid (GABA) neuron activation has been suggested to contribute to chronic stress-induced DA neuron hypofunction. VTA GABAergic neurons and their afferent inputs from diverse brain regions are highly responsive to stressful stimuli. 

A concurrent decrease in inhibition and increased excitation from the regions targeting VTA GABAergic neurons in stress can increase VTA GABAergic neuron firing [86]. Consequently, the exciting local VTA GABAergic interneurons could inhibit DA neurons through GABA_A_ receptor activation [87]. To observe the influence of stress on alcohol consumption, Alexey Ostroumov et al. used a single episode of restraint as a stressor and the ethanol self-management program. They noted the increase in extracellular dopamine concentration after alcohol intake in stressed mice was lower than that in the control group. Reduced mesolimbic dopamine release in response to alcohol diminishes the reward circuit [58]. Therefore, alcohol abusers require more alcohol to experience salience, promoting the development of dependence [88]. 

VTA GABAergic neurons can also be excited by aversive stimuli and functionally elicits negative emotions [86]. Negative moods lead PTSD patients to consume alcohol to relieve PTSD symptoms [89]. Notably, during the alcohol withdrawal period, PTSD-associated psychological symptoms such as anxiety and depression are persistent, which indicates that alcohol itself can provoke a stress response [90]. Thus, VTA GABAergic cells may be a potential target of PTSD and AUD comorbidity treatment. 

There is a strong association between alcohol addiction and the midbrain DA system. During the commencement of alcohol addiction, the VTA-DA neurons are activated, reflected by an increased in tonic and burst firings [55]. Whereas ablation of VTA GABAergic neurons increased ethanol intake [60], systemic administration of dopamine receptor antagonists can reduce cravings for alcohol in AUD patients [91]. The functional and structural changes of the VTA DA system promote/drive the VTA of AUD patients in a euphoric state and increases the risk of facing trauma. Brain-derived neurotrophic factor (BDNF) and glial cell line-derived neurotrophic factor (GDNF) is the most extensively studied neurotrophins in the VTA [92]. Short-term exposure to drug abuse increases BDNF levels in VTA neurons, promoting local DA neuron activity and positively reinforcing acute alcohol intake [93]. Likewise, GDNF is upregulated during short-term alcohol intake and exerts acute inhibitory effects on reducing alcohol consumption [94].

Chronic ethanol exposure produces neuroadaptations in DA circuits within VTA. These alterations, such as reduced dopamine receptors in NAc, remarkably decrease dopamine neuro-transmission in NAc and other related regions, suggesting that chronic ethanol consumption leads to DA hypofunction [95]. The above changes indicate the alterations drive excessive ethanol drinking [56]. The low VTA DAergic activity may impede the formation of fear-extinction memories [96], and fear and avoidance are core symptoms of PTSD. During alcohol withdrawal, profound decrement of VTA DAergic neuronal activity contributes to aversive or stress-like states [88], which can aggravate the negative mood and cognition symptoms in PTSD. Overall, alcohol addiction increases the possibility of being faced with trauma and may exacerbate PTSD symptoms.

### 3.2. Hippocampus

Hippocampus is characterized by its trisynaptic circuit architecture and is essentially involved in learning and memory, navigation, and cognition. Hippocampal dysfunction can lead to memory deficits, depression, epilepsy, and schizophrenia. PTSD and AUD are associated with profound changes in memory function and neuronal signaling related to the hippocampus. 

PTSD alters hippocampal neurons’ synaptic plasticity and firing properties, induces morphological atrophy, inhibits neuronal proliferation, and reduces hippocampal volume [54]. It impairs the function of the hippocampus and produces solid and long-lasting nociceptive memories. In addition, phosphorylation of mitogen-activated protein kinase (MAPK)/extracellular signal-regulated kinase (ERK) in the hippocampus has been proven to mediate the extinction of contextual freezing behavior, which is one of the main risk factors for PTSD [63].

Certain neuroimaging studies documented that acute alcohol exposure affects the episodic memory encoding function of the hippocampus [97]. Following the binge pattern of ethanol consumption, anaplastic lymphoma kinase signaling has been activated, and the transcription factor STAT3 level is increased in the brain [98]. There is also substantial evidence showing that AUD may be related to reduced hippocampal volume [99], decreased hippocampal neurogenesis [64,66], and downregulated mRNA expression levels [100].

Accumulating evidence has demonstrated that PTSD-AUD comorbidity is related to hypoxia, inflammation, and excessive cortisol secretion in the hippocampus [101,102]. Altered GABAergic receptor expression and reduced allopregnanolone levels have been identified in the comorbid mice compared to their littermates [65,103]. The above comorbid mice were induced by maternal separation or social isolation stress paradigm. Stress is one of the most commonly used methods to establish comorbidity models. Most recently, changes in the cannabinoid system enriched in the hippocampus are also reported to be implicated in comorbidity [104,105]. In this study, Veronica M. Piggott used chronic intermittent ethanol vapor exposure to increase the body ethanol concentration in mice, and weekly plasma samples were collected. The mice were intoxicated if the plasma ethanol concentration was above 175 mg/dL.

### 3.3. Amygdala

The amygdala has been proven involved in multiple physiological and behavioral responses to fear, stress, and substance use disorders [106,107]. Dysregulation of neuroplasticity in the amygdala has been recognized as one of the mechanisms in the pathophysiology of several mental illnesses, such as depressive and anxiety disorders [108]. 

As neuroimaging findings revealed, PTSD patients often display structural changes in the amygdala. For example, teenage or adult patients with relevant psychological symptoms both company with significantly lower amygdala volume, especially volumetric reductions of grey matter [109]. In addition, PTSD patients also suffer regional malfunction, which may depend upon different molecular mediators of plasticity, including glutamatergic NMDA-dependent mechanisms, BDNF, calcium-dependent mechanisms, and so on. Compared to healthy people, the amygdala of trauma patients shows more substantial activity [110]. This amygdala hyperactivity predicts blocked fear extinction in the brain, while low amygdala activity predicts impaired response to fear restimulation [111]. Therefore, the hyperactivity of the amygdala elicited by negative stimuli may be indicative of trauma pre-traumatic events.

AUD patients had smaller amygdala volumes, which were positively associated with anxiety and negative urgency in AUD [112]. In particular, the GABAergic neurotransmitter system and adrenergic receptors in the central amygdala (CeA) have been implicated in regulating acute and chronic drinking behaviors [113]. Specifically, acute ethanol exposure activates α1 receptors and potentiates CeA GABAergic transmission, while chronic alcohol consumption activates β receptors and disinhibits a subpopulation of CeA neurons leading to their sustained hyperactivity [68,114]. The action of CRHR1 in the CeA was enhanced by ethanol exposure, suggesting that the CRH signaling pathway affects both the pre and postsynaptic transmission of GABA. Mineralocorticoid receptors (MR) in the CeA have been found to modulate alcohol self-administration and showed inverse correlations between its expression and measures of alcohol drinking [115,116]. 

In terms of the comorbidity, a 2-hit model was proposed to examine increased GABAergic transmission, and expressions of different neuroinflammatory factors, including G-CSF, and IL-13, were profiled in the amygdala [117,118]. Clinical studies have also demonstrated higher amygdala blood flow in AUD and PTSD patients than in healthy controls [119]. Despite the significantly lower hippocampal volume observed in PTSD and AUD patients, other brain areas appear not to be different by volumetric analysis [120]. 

### 3.4. PFC

The PFC is a portion of the cerebral cortex covering the major part of the frontal lobe. The ventromedial prefrontal cortex (vmPFC) and dorsolateral prefrontal cortex (dlPFC) are closely associated with social, cognitive, and affective functions. 

PTSD patients often showed morphological defects, such as a lower volume [109], especially in the gray matter [121] of the vmPFC, accompanied by reduced regional cerebral blood flow [122]. The interconnectivity between the PFC and the amygdala [123] suggests that the enhanced emotional memory traces in PTSD patients may result from an unbalanced interaction between two brain regions and PFC inhibition [124]. Additionally, reduced connectivity in dlPFC and medial PFC of PTSD cases have been uncovered through fMRI studies [125]. It has also been suggested that dephosphorylation of the mechanistic target of rapamycin (mTOR) and its upstream kinase, protein kinase B (Akt), in the PFC results in the disappearance of PTSD symptoms like fear extinction [126]. Thus, PTSD is associated with PFC hypofunction.

Structural and functional MRI studies suggest that disruption to the circuits originating from the PFC plays a crucial role in cognitive and motor impairments in AUD patients [127]. AUD is associated with reduced gray matter volume of corticostriatal and marginal circuit components such as dlPFC, which contributes to regional executive dysfunction. Reduced inhibition and enhanced excitatory synaptic activity of the PFC in addiction are well-documented in fMRI studies and are associated with increased adverse clinical outcomes [82,128]. In addition, the role and impact of the PFC-striatum circuit [129] and the PFC-amygdala circuit have been under intensive investigation recently [130].

Functional and structural alterations are observed in the prelimbic region of the PTSD and comorbid AUD patients’ PFC, which is especially important due to its involvement in promoting and suppressing fear and ethanol-seeking behavior [43]. It is noted that these deficits could be treated by enhanced activity at metabotropic glutamate receptor 5 (mGluR5) in this region [70]. Using optogenetics to suppress prelimbic activity, we found that fear memory reconsolidation and addiction behaviors were blocked [81]. Likely, PTSD patients with alcohol abuse showed lower blood flow in the PFC [119]. 

### 3.5. PVN

PVN is an integral part of the hypothalamic-pituitary-adrenal (HPA) axis, composed of three distinct anatomical loci: the PVN of the hypothalamus, the pituitary gland, and the adrenal cortices. The stress system, comprising of the HPA axis, and the locus (LC)/norepinephrine (NE)-autonomic nervous system, can be activated by stressful stimuli and initiates central and peripheral neuroendocrine responses to maintain regular homeostasis [131]. However, aberrant responses may lead to PTSD and AUD.

Trauma has been evidenced to contribute to HPA axis disturbances [132]. Glucocorticoids can easily cross the blood–brain barrier and produce negative feedback on the HPA axis, thereby reducing the secretion of corticotropin-releasing hormone (CRH) and adrenocorticotrophic hormone (ACTH). Under the circumstance of trauma, the secretion of hypothalamic CRH elevates rapidly, leading to enhanced release of ACTH [133]. The glucocorticoid secretion occurs more slowly [134]. The HPA axis activates the CeA, which generates negative emotions, such as fear and anger. The CeA stimulates the stress system, producing a positive regulatory feedback loop [135]. People with PTSD showed a flat distribution of cortisol levels throughout the day and night [136], which indicates that PTSD patients are more vulnerable to an arousal state.

AUD patients also show changes in HPA axis function, as the HPA axis has been suggested to play an essential role in drinking behavior [137]. Alcohol initially increases HPA axis activity leading to autonomic arousal, which further potentiates alcohol-related striatal transmission to promote motivation and rewarding properties of alcohol [138,139]. Alcohol also increases the NAc dopamine release [140], so glucocorticoids may regulate drinking behavior by acting on the reward system of the limbic brain, which expresses a large amount of GR [141]. However, heavy and excessive alcohol consumption leads to adaptations and debilitation to neuroendocrine regulation circuits and the reward system [142], which results in the transition from controlled to compulsive drinking behavior [143]. In the withdrawal of alcohol consumption, the negative reinforcement may drive AUD patients to intake more alcohol. Relapse-associated resumed consumption of alcohol would worsen the HPA axis and autonomic dysfunction, which hinders recovery from alcohol addiction [144].

### 3.6. LC

The LC in the central NE system plays an irreplaceable role in daily stress response and fear response. The LC is a cluster of NE-synthesizing neurons in the pontine brainstem adjacent to the fourth ventricle, which projects widely throughout the entire neuraxis [145]. LC-NE, along with other components, constitutes the stress system, which affects several psychological processes, including arousal, attention, and some control of fundamental physiological processes such as emotion regulation and cognition.

One of the significant symptoms of PTSD patients is flashing out, which is shown as cognition, memory, and arousal deficits. LC-NE is involved in stress responses and regulating cognitive function via PFC [146]. Acute stress increases tonic LC activity to facilitate alertness and scanning attention [146]. However, exposure to prolonged, chronic stress would cause LC-NE dysfunction [147], which can worsen PTSD symptoms. This statement is supported by a finding of aggravated core symptoms (including anxiety and avoidance) in people with PTSD using an α2-adrenergic receptor antagonist [148].

NE is involved in multiple aspects of motivation, including compulsive behaviors associated with alcohol abuse [149]. There is ample evidence that acute and chronic drinking can affect NE neuronal function and NE release [150]. Stressors have been documented to reinstate drug-seeking behavior [149,151], and the NE system is activated by stress and exerts robust arousal-promoting actions. The stress of withdrawal acts as a negative reward, encouraging AUD patients to relapse.

## 4. Conclusions

Similar anatomical and functional changes observed in the overlapping brain regions and the high incidence of comorbidities between AUD and PTSD suggest that these two etiologically distinct disorders share specific common neurobiological mechanisms in their pathogenesis (summerized in Figure 1). Since the activity of a brain area interacts with and affects other brain areas via mutually connected pathways, investigating the comorbidity is challenging. This review focuses on the structural and functional changes in corresponding brain regions in one disease, followed by the dysfunction of the shared neural circuits in another condition.

A literature search for studies about the individual neurobiology of PTSD and AUD returns many results, with much less on their comorbidities. It appears that PTSD and AUD coexist in a feed-forward process in which PTSD symptoms lead to excessive alcohol consumption, and binge drinking worsens PTSD symptoms.

Additionally, it has been suggested that early exposure to stress or alcohol in childhood may increase the risk of AUD and PTSD in adulthood. At the longitudinal level, it remains unclear whether there are critical periods, such as adolescence, during which individuals are particularly vulnerable to traumatic stress and alcohol addiction. In addition, the theoretical hypothesis of comorbidity has been analyzed chiefly based on existing small samples, lacking laboratory animal model data for verification, limiting the possibility of excluding influential factors in the analysis process due to the insufficient sample size.

Moreover, developing an effective and translational comorbid animal model is not trivial. Preclinical studies regarding comorbidity have been challenging and complicated because of the varying intensity, duration, and types of stress superimposed on different brain regions, circuits, neurotransmitters, and individual differences within a model. Behavioral symptoms and self-reports are used to diagnose comorbidity, but there are no objective parameters, and there is significant overlap with other mental illnesses. Therefore, novel technologies such as chemo-optogenetics should be supplemented based on adopting traditional experimental protocols.

## Figures and Tables

**Figure 1 ijms-24-00388-f001:**
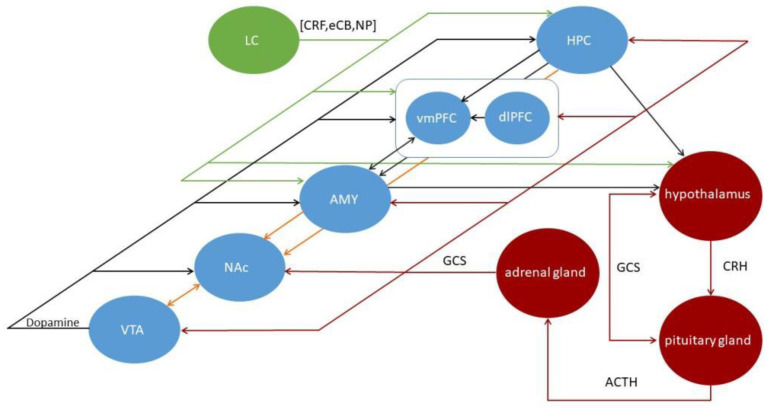
The illustration of the predominant brain regions and circuits is generally thought to mediate the processing of reward (orange), HPA (red), and NE system (green) in the human brain. The NAc is usually the central node of reward circuits, including the DAe input from the VTA and excitatory input from other limbic structures. CRH from the hypothalamus stimulates ACTH release from the anterior pituitary into the bloodstream, then ACTH induces GCS release from the adrenal gland. GCS mediates negative feedback in the HPA axis to reduce the stress response. GCS also affects the circuitry via the GCS receptors, triggering molecular and cellular. The primary stress-related central NE system originates in the LC and plays an integral role in the comorbidity. Some of the many connections that link elements of the comorbidity are illustrated by black lines. LC, locus coeruleus; CRF, corticotropin-releasing factor; eCB, endocannabinoid; NP, neuropeptide; HPC, hippocampus; vmPFC, ventromedial prefrontal cortex; dlPFC, dorsolateral prefrontal cortex; AMY, amygdala; NAc, nucleus accumbens; VTA, ventral tegmental area; GCS, glucocorticoids; CRH, corticotropin-releasing hormone; ACTH, adrenocorticotrophic hormone; NE, norepinephrine.

**Table 1 ijms-24-00388-t001:** The common paradigms and the number of studies per each paradigm in rodent PTSD studies.

Common Models	Before 2010 (–31 December 2009)		Before 2020 (1 January 2010–31 December 2019)		Until the End of 2022 (1 January 2020–2022)	
	Rats	Mice	Rats	Mice	Rats	Mice
Restraint/immobilization stress	48	4	101	43	33	18
Underwater trauma	4	0	9	1	5	2
Single prolonged stress	22	1	212	20	93	30
Electric shock	10	4	40	30	9	6
Social defeat	1	1	16	21	8	13
Predator stress	33	9	95	19	35	10

PubMed search with the keywords “PTSD” or “post-traumatic stress disorder” and the name of the stress paradigm and “rat” or “mice” and “Date of publication” was conducted in November 2022. The digit in the table indicates the number of studies in PTSD studies.

**Table 2 ijms-24-00388-t002:** Summary of tests used in animal models of PTSD.

	Cluster B: Intrusion	Cluster C: Avoidance	Cluster D: Negative Alterations in Mood and Cognition	Cluster E: Alterations in Arousal and Reactivity
Test + *Key parameter*	Contextual and cued trauma reminders (*freezing: in trauma context or fear generalization*)	Open field (*reduced centrum/open arm time*)	Morris water maze/Barnes maze (*spatial memory disturbance*)	Electroencephalogram (*Increased rapid eye movement (REM), sleep fragmentation*)
		Elevated T/zero/plus-maze (*reduced centrum/open arm time*)	Radial arm water maze (*spatial memory disturbance*)	Prepulse inhibition (*abnormal sensorimotor gating*)
		Light-dark box (*reduced time spent in light*)	Anhedonia (*reduced sucrose preference*)	Operant attentional set-shifting tasks (*number of errors to criterion*)
		Conditioned odor active avoidance (*reduced time spent in acetic acid compartment*)	Forced swim (*increased immobility*)	Aggression (*aggressive outburst*)
		Object/marble burying (*increased burying*)	Novel object recognition (*decreased discrimination abilities*)	Object/marble burying (*enhanced number of buried marbles*)
		Object exploration (*reduced object exploration*)	Social preference/avoidance test (*reduced sucrose preference*)	Startle response (*enhanced startle response and reactivity*)
			Response bias probabilistic reward task (*response bias and discriminability to rewards*)	Open field (*hyperlocomotion*)
			Step-through inhibitory avoidance (*increased latency time to enter the dark compartment*)	

**Table 3 ijms-24-00388-t003:** Summary of tests used in animal models of AUD.

	Behavioral Sensitization	Rewarding Property	Ethanol Tolerance	Aversion
**Binge drinking**	Open field (*increased centrum/open arm time*)	CPP *(time spent in different places)*	LORR *(time of occurrence, alcohol concentration)*	CPA *(time spent in different places)*
			Hypothermia test *(tolerance time)*	CTA *(median saccharin preferences scores)*
**Withdrawal-associated negative affective disorders**	**Anxiety-like behavior**	**Anhedonia/depression**		
	Object/marble burying (*enhanced number of buried marbles*)	Reduced sucrose preference		
	Elevated T/zero/plus-maze (*reduced centrum/open arm time*)	Forced swim (*increased immobility*)		
**Craving behaviors**	**Ethanol consumption**	**Ethanol seeking motivation**		
	Drinking in dark *(EtOH intake)*	SA *(EtOH intake)*		
	Two bottle choices *(EtOH intake, preference for EtOH)*	Rewarding EtOH-lever press *(number of pressing)*		
	IA2BC *(EtOH intake, preference for EtOH)*	Punishment-imposed EtOH-lever press *(“break point”)*		

Note: CPP, conditioned place preference; LORR, loss of the righting reflex; CPA, conditioned place aversion; CTA, conditioned taste aversion; SA, self-administration; IA2BC, intermittent access to ethanol under a 2-bottle choice; Median saccharin preferences scores = (saccharin solution intake/total fluid intake) × 100.

**Table 4 ijms-24-00388-t004:** Preclinical models and circuitry of PTSD or/and AUD.

	PTSD		AUD		PTSD&AUD	
	Molecular Changes	Preclinical Models	Molecular Changes	Preclinical Models	Molecular Changes	Preclinical Models
**VTA**	DA↑	RS, ES, SDS, etc. [53,54]	DA↑↓	IA2BC [55], SA [56]	DA↑	Adolesent social isolation [57]
	GABA↑↓	RS, SDS, etc. [58,59]	GABA↓	IA2BC [60]	DA↓(in NAc)	Social isolation [61]
	Glu↑↓	SDS, RS, US, etc. [48,62]				
**Hippocampus**	Phosphorylation of MAPK/ERK	ES [63]	Decreased neurogenesis	SA [64]	GABAR↑	Adolesent social isolation [65]
			Reduced neuroplasticity	SA [64]	LTP↓	Social isolation [66]
**Amygdala**	Fear extinction↑; frequency of nightmares↓	DBS in Macaca fascicularis [67]	GABA↑	Alcohol vapor [68]	CRF↑	Predator odor [69]
**PFC**	Dephosphorylation of mTOR and Akt	ES (conditioned/unconditioned stimulus) [70]	Fos (+); neurons↑(withdrawal)	2BC/IA2BC [71]	Metabotropic glutamate receptor 5↓	Fear conditioning + alcohol vapor [70]
**PVN**	HPA axis activation↓	Predator odor [72]	HPA function↓	SA [73]	Stress peptide CRH↑	Maternal deprivation [74]
**LC**	β-adrenoceptor mediated signaling ↑for memory reconsolidation	SA [75]	α1-adrenergic receptor↑	SA + alcohol vapor [76]	NE neurotransmission↑	Adolesent social isolation [77]

Note: DA, dopamine; GABA, gamma-aminobutyric acid; Glu, glutamate; RS, restraint stress; ES, electric stress; SDS, social defeat stress; IA2BC, intermittent access to ethanol under a 2-bottle choice; NAc, nucleus accumbens; MAPK, mitogen-activated protein kinase; Erk, extracellular signal-regulated kinases; SA, self-administration; US, uncontrollable stress; LTP, long-term potentiation; DBS, deep brain stimulation; CRF, corticotropin-releasing factor; HPA, hypothalamic-pituitary-adrenal; NE, norepinephrine; Arrows indicate the changing trend of molecules.

## Data Availability

Not applicable.

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
