# Peer review of "Comorbidity of Post-Traumatic Stress Disorder and Alcohol Use Disorder: Animal Models and Associated Neurocircuitry"

_ijms, 2022, doi:10.3390/ijms24010388_

Round 1

Reviewer 1 Report

The writing is confusing and extensive revision is needed. The descriptions of both PTSD and AUD are odd, perhaps because of the writing style. It is commonly accepted that AUD is a separate disorder, requiring separate treatment, no matter what other disorder with which it is comorbid. Overall, the paper takes a great deal of information from others and attempts to turn it  into a guideline of the brain regions, neurotransmitters, and brain activity subserving the two disorders. The paper ends up being too confusing to be of any value.

Author Response

  1. The writing is confusing and extensive revision is needed.

Response:

The manuscript was extensively revised and edited.

We rewrote all the confusing texts to make the revision version much more readable and clear.

  1. The descriptions of both PTSD and AUD are odd, perhaps because of the writing style.

Response:

We rewrote the description of PTSD and AUD in the introduction section.

Please review the revised manuscript's lines 43-56 as follows:

“PTSD is a maladaptive and debilitating psychiatric disorder characterized by re-experiencing, avoidance, negative emotions and thoughts, and hyperarousal in the months and years following exposure to severe traumatic events [1]. According to DSM-5 diagnostic criteria, PTSD patients are typically subject to the traumatic event (Criteria A) accompanied by following core symptoms over one month: 1) Intrusive memories as Criteria B. 2) Avoidance behaviors as Criteria C.  3) Negative mood and cognition as Criteria  D. 4) Arousal and sleep disturbances as Criteria E [2].  Alcohol use disorder (AUD) is a chronic brain disease involving the transition from recreational drinking to dependence, characterized by a negative affective state, excessive alcohol intake, and constant motivation to consume alcohol despite adverse consequences. AUD involves binge or heavy drinking to high levels of intoxication, leading to compulsive intake, losing control in limiting intake, and a negative emotional state when alcohol is removed[3,4]. A strong correlation between PTSD and alcohol misuse has yet to be established [5]. The interaction between alcohol and PTSD is often reciprocal.

  1. It is commonly accepted that AUD is a separate disorder, requiring separate treatment, no matter what other disorder with which it is comorbid.

Response:

We agree with the reviewer’s opinion that AUD requires different treatments for PTSD.

Indeed, as we mentioned in the introduction, the comorbidity of PTSD and AUD has been well documented by numerous preclinical and clinical evidence. Notably, the two disorders affect each other during the process and treatment, challenging how we manage patients using traditional treatment protocols. Therefore, it urges the researchers to investigate the interaction between two psychiatric disorders and find common underpinning mechanisms at the neural circuitry and molecular level.

  1. Overall, the paper takes a great deal of information from others and attempts to turn it into a guideline of the brain regions, neurotransmitters, and brain activity subserving the two disorders. The paper ends up being too confusing to be of any value.

Response:

Thanks for pointing out the weakness of the current article.

Since the mechanism studies regarding comorbidities of AUD and PTSD are still lacking, the main aim of the current review article is to try to help readers in the field to summarize the research advances in neural adaptation involved PTSD and AUD respectively, providing the necessary trend, direction, and discussion for future comorbidity study.  

We reorganize the information from animal studies published within the past ten years into 4 new tables to make the manuscript more visual and understandable.

Table 1 summarizes the prevalence of PTSD rodent models. New tables were added to list the commonly used behavioral assays for PTSD (Table 2) and AUD (Table 3) model validity. Table 4 illustrates neuron adaptation changes in certain key brain regions involved in PTSD or/and AUD.

Please review the new Tables 1-4 in the revised manuscript.

Reviewer 2 Report

Bo Zhan et al. submitted a review article, titled in “Comorbidity of post-traumatic stress disorder and alcohol use disorder: a review of preclinical models and neurobiology”. This article discusses how post-traumatic stress disorder (PTSD) and alcohol use disorder (AUD) are correlated with each other in the introduction part, and suggests the animal models for research in the PTSD-AUD comorbidity. Additionally, the article describes the neurobiological mechanisms of the disease in each different brain region. There are several points I suggest although the article provides useful information for the comorbidity of the PTSD and AUD disorder.

1.       Part 2 should be detaily explained for better understanding.

(1)    On line 103-104, detailed information about the face validity, construct validity and predictive validity in the PTSD or AUD research should be described.

(2)    More detailed behavioral tests should be displayed. What is the reason to perform these behavioral tests especially for PTSD-AUD comorbidity? Which tests are for PTSD and AUD? How can the PTSD-AUD comorbidity can be determined using these methods? Do you have any idea or suggestion how the tests should be combined or modified to explain the relationship between PTSD and AUD rather than the single disorder? Each category should be separated with the subtitles.  

(3)     Animal models should be more described. What is each PTSD and AUD model the previous studies used? How do you prepare the animal for the PTSD-AUD comorbidity? Mouse, rat, monkey, and other animal models, and their aging, sex dependent effects as well as any genetic effect should be described detailly. Switching the order of the animal model part with the behavior test part is considerable if it is helpful for better understanding. Tables showing animal models and behavior tests will be helpful too.

2.       Part 4, 5, 6, 7, 8, and 9 should belong to part 3 as subtitles. The term “Neurobiology” seems vague. I suggest removing the “neurobiology” from title of the part 3 as well as from the main title and using “neurocircuitry” instead.

3.       The parts of preclinical models and the neurocircuitry are too separated from each other. What kind of animal model or the behavioral test was used to determine the brain regions or certain types of neurons in the PTSD-AUD comorbidity? The relationship between animal models and found brain regions should be described in this part. Making a table will be helpful too.       

Author Response

  1. Part 2 should be detailly explained for better understanding.

(1)    On line 103-104, detailed information about the face validity, construct validity and predictive validity in the PTSD or AUD research should be described.

Response:

Part 2 has been extensively revised as suggested.

New Tables 1-3 were added to summarize detailed information (like key parameters and experimental aim) regarding establishing PTSD and AUD rodent models and their behavioral validity methods.

Detailed information about the face validity, construct validity, and predictive validity in the PTSD or AUD research was added in lines 144-151 as follows:

“Three criteria are frequently induced to assess the eligibility of PTSD animal models: face validity, construct validity, and predictive validity [6]. Face validity means the animal model should show similar PTSD-like symptoms to those detected in humans, especially the four core symptoms listed above. Construct validity is a theoretical rationale. Since the cause (traumatic experiences) is relatively straightforward, a model with a similar construct (using trauma exposure) can be developed. Moreover, predictive validity shows whether the model can predict performance, in which how well the test works is established by measuring it against known criteria.”

Please also review the new Table 1, Table 2, and Table 3.

(2.1)    More detailed behavioral tests should be displayed.

Response:

Detailed behavioral tests have been displayed as suggested.

Please review the information in Part 2.

(2.2)   What is the reason to perform these behavioral tests, especially for PTSD-AUD comorbidity? Which tests are for PTSD and AUD?

Response:

Since the animal model has sought to mimic the core symptoms of PTSD and AUD, as mentioned in lines 45-49 and lines 49-54 in the introduction section, the reason for performing behavior tests is to assess the phenotypes and validate the established animal model, especially for PTSD-AUD comorbidity.

The aim of each test for PTSD and AUD was well described and listed in the new Tables 2 and 3.

(2.3)    How can the PTSD-AUD comorbidity be determined using these methods?

Response:

We added new references to explicate how the PTSD-AUD comorbidity can be determined by using these methods, which have been provided in lines 274-287:

“Generally, testing methods used in animal studies addressing comorbidity are based on experimental aims or interests. It’s expected to determine whether the comorbidity model is established according to how these two disorders couple. For example, McCool BA et al. made the rats subjected to social isolation for 6 weeks, validated by elevated plus maze and light/dark box test, to ensure anxiety symptoms expression, mimicking PTSD. Then, to investigate how PTSD affects subjects' responses to alcohol-rewarding properties, these rats continued to receive ethanol-induced CPP test and alcohol consumption test under the 2BC drinking paradigm. Interestingly, the authors found that PTSD produced long-lasting increases in ethanol self-administration and ethanol-conditioned place preference[7]. Notably, however, it has also been claimed that excessive drinking increases the risk of developing PTSD. For example, Holmes and colleagues demonstrated that mice intermittently exposed to continuous vaporized alcohol and withdrawal were likely to obtain impaired fear extinction, which means a reduction in the frequency or intensity of a conditioned fear response (e.g., freezing) [8].”

(2.4)    Do you have any idea or suggestion on how the tests should be combined or modified to explain the relationship between PTSD and AUD rather than the single disorder?

Response:

Yes, the suggestions were provided in the discussion of the Part 2 section. Please review the added contents in lines 274-287.

(2.5)    Each category should be separated with subtitles.  

Response:

       Done as suggested.

(3.1)     Animal models should be more described. What is each PTSD and AUD model the previous studies used? How do you prepare the animal for the PTSD-AUD comorbidity?

Response:

       Detailed information about the animal models has been added as suggested.

We provided information regarding each PTSD and AUD model used in the previous studies in lines 123-143 and 159-184.

In addition, we explicated the way to prepare the animal for the PTSD-AUD comorbidity in lines 254-273.

(3.3)     Mouse, rat, monkey, and other animal models, and their aging, sex dependent effects as well as any genetic effect should be described detailly.

Response:

Besides the rodent models, we added other species models used in PTSD and AUD respectively, in lines 113-116 and 155-158 as follows:

“And in some specific studies, rabbits, voles, or zebrafish were also utilized. Caution is needed when interpreting results since significant differences regarding morphological structures, stress responses, physiologies, genetics, and behaviors exisit between the two species [9].”

“Essential characteristics of alcohol consumption have been found in many mammal species, fish, birds, and non-spinal animals[10]. However, significant neuropharmacology and brain circuitry differences between non-mammalian model systems and mammals limit the use of animals mentioned above. Rodents are the most commonly used animal models.”

Aging, sex-dependent, and genetic effects have been described in lines 116-121 and 223-228.

“Moreover, over the past two decades, more mice were utilized in studies to dissect mechanisms related to immunology or genetics (Table 1). It’s worth noting that epidemiological assessments of gender have emphasized the importance of studying the susceptibility of females, female traumas, and sexual abuse. For example, in rats exposed to predators’ odor, females were more likely to exhibit PTSD-like phenotype, while female models were hard to establish when exposed to social defeat[11].”

“Animal models of AUD have limitations. Firstly, it is necessary to reduce the influence of animals’ susceptibility to alcohol to develop alcohol addiction models more efficiently. Additionally, AUD symptoms include mental and physical symptoms, but in animal models, we can only observe mental changes through behavioral tests, which limits the overall understanding of the disease. In addition, sex and species differences also should not be ignored.”

(3.4)     Switching the order of the animal model part with the behavior test part is considerable if it is helpful for better understanding.

Response:

       Thanks for the constructive suggestions.

The two parts have been switched as suggested.

(3.5)     Tables showing animal models and behavior tests will be helpful too.

Response:

       We have added the brand-new Table 1, 2, and 3 to illustrate animal models and behavioral tests.

  1. Part 4, 5, 6, 7, 8, and 9 should belong to part 3 as subtitles. The term “Neurobiology” seems vague. I suggest removing the “neurobiology” from the title of the part 3 as well as from the main title and using “neurocircuitry” instead.

Response:

The manuscript and the title were revised as suggested.

  1. The parts of preclinical models and the neurocircuitry are too separated from each other. What kind of animal model or the behavioral test was used to determine the brain regions or certain types of neurons in the PTSD-AUD comorbidity? The relationship between animal models and found brain regions should be described in this part. Making a table will be helpful too. 

Response:

       We appreciate the reviewer’s thoughtful and constructive comments. New Table 4 was added to connect the two parts to cover the specific animal model or the behavioral test used in particular brain regions or certain types of neurons of the PTSD-AUD comorbidity.
